# Evaluation of the Usefulness of Human Adipose-Derived Stem Cell Spheroids Formed Using SphereRing^®^ and the Lethal Damage Sensitivity to Synovial Fluid In Vitro

**DOI:** 10.3390/cells11030337

**Published:** 2022-01-20

**Authors:** Atsushi Fuku, Yasuhiko Taki, Yuka Nakamura, Hironori Kitajima, Takashi Takaki, Terutsugu Koya, Ikuhiro Tanida, Kaori Nozaki, Hiroshi Sunami, Hiroaki Hirata, Yoshiyuki Tachi, Togen Masauji, Naoki Yamamoto, Yasuhito Ishigaki, Shigetaka Shimodaira, Yusuke Shimizu, Toru Ichiseki, Ayumi Kaneuji, Satoshi Osawa, Norio Kawahara

**Affiliations:** 1Department of Orthopedic Surgery, Kanazawa Medical University, Kahoku 920-0293, Ishikawa, Japan; f-29@kanazawa-med.ac.jp (A.F.); taki8468@kanazawa-med.ac.jp (Y.T.); kuppy@kanazawa-med.ac.jp (H.K.); hiro6246@kanazawa-med.ac.jp (H.H.); y-t-s17@kanazawa-med.ac.jp (Y.T.); kaneuji@kanazawa-med.ac.jp (A.K.); kawa@kanazawa-med.ac.jp (N.K.); 2Medical Research Institute, Kanazawa Medical University, Kahoku 920-0293, Ishikawa, Japan; yuka-n@kanazawa-med.ac.jp (Y.N.); ishigaki@kanazawa-med.ac.jp (Y.I.); 3Division of Electron Microscopy, Showa University, Tokyo 142-8555, Japan; takaki@med.showa-u.ac.jp; 4Center for Regenerative Medicine, Kanazawa Medical University Hospital, Kahoku 920-0293, Ishikawa, Japan; koya@kanazawa-med.ac.jp (T.K.); shimodai@kanazawa-med.ac.jp (S.S.); 5Department of Regenerative Medicine, Kanazawa Medical University, Kahoku 920-0293, Ishikawa, Japan; 6Genome Biotechnology Laboratory, Kanazawa Institute of Technology, Hakusan 924-0838, Ishikawa, Japan; tanida@neptune.kanazawa-it.ac.jp (I.T.); osawa@neptune.kanazawa-it.ac.jp (S.O.); 7Department of Pharmacy, Kanazawa Medical University Hospital, Kahoku 920-0293, Ishikawa, Japan; knozaki@kanazawa-med.ac.jp (K.N.); masauji@kanazawa-med.ac.jp (T.M.); 8Advanced Medical Research Center, Faculty of Medicine, University of the Ryukyus, 207 Uehara, Nakagami 903-0215, Okinawa, Japan; sunami@med.u-ryukyu.ac.jp; 9Center for Clinical Trial and Research Support, Research Promotion and Support Headquarters, Fujita Health University, Toyoake 470-1192, Aichi, Japan; naokiy@fujita-hu.ac.jp; 10Department of Ophthalmology, Kanazawa Medical University, Kahoku 920-0293, Ishikawa, Japan; 11Department of Plastic and Reconstructive Surgery, Graduate School of Medicine, University of the Ryukyus, Nakagami 903-0215, Okinawa, Japan; yyssprs@gmail.com

**Keywords:** spheroid, adipose-derived stem cell, synovial fluid, knee osteoarthritis, cell methods

## Abstract

Osteoarthritis (OA) is an irreversible degenerative condition causing bone deformation in the joints and articular cartilage degeneration with chronic pain and impaired movement. Adipose-derived stem cell (ADSC) or crushed adipose tissue injection into the joint cavity reportedly improve knee function and symptoms, including pain. Stem cell spheroids may be promising treatment options due to their anti-inflammatory and enhanced tissue regeneration/repair effects. Herein, to form human ADSC spheroids, we used first SphereRing^®^ (Fukoku Co., Ltd., Ageo, Japan), a newly developed rotating donut-shaped tube and determined their characteristics by DNA microarray of mRNA analysis. The variable gene expression cluster was then identified and validated by RT-PCR. Gene expression fluctuations were observed, such as COL15A1 and ANGPTL2, related to vascular endothelial cells and angiogenesis, and TNC, involved in tissue formation. In addition, multiplex cytokine analysis in the medium revealed significant cytokines and growth factors production increase of IL-6, IL-10, etc. However, ADSC administration into the joint cavity involves their contact with the synovial fluid (SF). Therefore, we examined how SF collected from OA patient joint cavities affect 2D-culture ADSCs and ADSC spheroids and observed SF induced cell death. ADSC spheroids could become promising OA treatment options, although studying the administration methods and consider their interaction with SF is essential.

## 1. Introduction

Osteoarthritis (OA) is an irreversible degenerative condition that causes deformation of the bones that make up the joints, as well as degeneration of the articular cartilage, causing chronic pain and impaired movement of the extremities. Inflammation of the synovial membrane surrounding the joints also occurs, resulting in its degeneration. This may be accompanied by proliferative changes, such as osteochondrogenesis, around the joints, and these changes are thought to cause fibrosis of the joint capsule accompanied by angiogenesis and nerve fiber proliferation, leading to pain [1,2,3]. The prevalence of OA is 10% in males aged >45 years and 18% in females aged >45 years, and the rate of degeneration determined using X-ray is 80% at 65 years. OA reportedly affects around 250 million people worldwide [4,5,6]. More than 80% of OA patients complain of restricted joint movements, and more than 25% are unable to perform normal activities in daily activities. Therefore, OA is considered a serious disease that is associated with great socioeconomic impact and a heavy burden on individuals [7,8]. Aging and obesity are risk factors for its onset and progression; therefore, the impact of OA on society is expected to increase in the future [9,10,11].

The frequency of knee OA is high and affects many patients; however, symptomatic treatment remains the main treatment. Lifestyle guidance, equipment therapy, rehabilitation, drug therapy (nonsteroidal anti-inflammatory drugs, pregabalin, tramadol), injection therapy (steroids, hyaluronic acid (HA)) are performed as conservative treatment, although their effectiveness is limited and cannot prevent the progression of degeneration [12,13]. Total knee replacement is selected when conservative treatment is uncontrollable using pain surgical treatments, such as arthroscopy and osteotomy. There is currently no well-established treatment other than total joint replacement [14].

Conservative treatment resistance is common, and the development of a treatment method for joints is required. In addition, some patients do not wish to undergo surgery, and treatments are required to fill the gaps between conservative and surgical treatments. In recent years, cell therapy involving intra-articular administration, such as platelet-rich plasma and mesenchymal stem cells (MSCs), has been performed for OA patients in early to advanced stages [15,16,17].

MSCs are stem cells derived from mesodermal tissues, such as bone marrow and fat, that can differentiate into various tissues, such as bone, cartilage, muscle, fat, and tendon, and control immune function [18,19]. Among them, adipose tissue-derived stem cells (ADSCs) are abundant in adipose tissue and can be easily collected in large quantities. Around 100 to 1000 times more MSCs can be collected from adipose tissue than from the same amount of bone marrow tissue; therefore, adipose tissue is an important source of MSCs [20]. ADSCs have a high immunoregulatory function since they secrete many immunomodulators and growth factors, such as IL-6, IL-10 and VEGF and so on [6,21]. These cytokines are thought to play an important role in therapeutic efficacy. Injection of cultured ADSCs or crushed adipose tissue into the joint cavity improves knee function, clinical symptoms, such as pain, and delays the progression of OA [22,23,24,25,26,27,28].

The term spheroid refers to a cell aggregate in which cells adhere to each other in three-dimensional culture. Gene expression in spheroids is thought to be maintained over a long period of time [29]. In recent years, spheroids obtained by three-dimensional culture have attracted attention in the field of cell therapy. The use of spheroids is expected to achieve more effective treatment by reproducing a state that is close to that of the microenvironment in the living body [30,31,32,33].

The advantages of MSC spheroid formation include anti-inflammatory effects, anti-apoptotic effects, enhancement of tissue regeneration/repair effects by promoting angiogenesis, homing to the injury site, promotion of antitumor factors, and anti-inflammatory factor paracrine secretion. MSC spheroid formation may improve cell viability, enhance differentiation potential, delay replication aging of MSCs, and be used to enhance certain therapeutic effectiveness [34,35,36,37]. This makes spheroids using ADSCs good candidates for the treatment of OA.

Spheroid formation using a culture technique that can purify a large number of spheroids of a uniform size is desired. Several culture methods aimed at clinical application have been reported, including a method of seeding cells on a low-adhesion plate and swirling lightly, a method using a 96-well plate, a non-adherent specific dish method, a method using a spinner flask (bioreactor method), a hanging drops method, and a rotary cell culture system [38,39,40].

The recently developed SphereRing^®^ is a ring-shaped culture vessel for forming spheroids, which is swirled on a shaker to produce spheroids from floating cells. A simple circular bag would form a large spheroid in the center when swirled, but the donut shape prevents accumulation in the center and allows for the formation of multiple spheroids of smaller uniform size. Compared to previous culture vessels, the advantage of SphereRing is that large amounts of spheroids can be prepared for clinical treatment. These spheroids can be easily produced, and closed-type culture can be performed because it uses a highly gas-permeable film. However, it is difficult to uniformly adjust the size of the spheroids, and it is necessary to set up the appropriate conditions in advance [41]. SphereRing has been used for various cell types, such as iPS, HEK293, and CHO cells, but no trial has been performed yet with ADSCs.

The joint cavity is filled with joint fluid, which is expected to have a significant effect on ADSCs administered intra-articularly in clinical treatments. It is important to consider the effect of synovial fluid (SF) on the survival rate of ADSCs when considering their therapeutic effect, although there have been few to date [42]. Kiefer et al. reported that SF from patients with knee OA may be cytotoxic to ADSCs administered intra-articularly [43]. Therefore, it is important to consider the effects that human SF within the joint cavity has on administered human ADSCs.

In the present study, we performed the first human ADSC spheroid formation experiment using SphereRing. We successfully formed spheroids using ADSCs and established the optimum formation conditions. We then clarified changes in the transcriptome and multiple cytokine production in the prepared spheroids compared with that in monolayer cultured cells. Finally, we examined the effects of adding SF to the culture medium and compared the effects on ADSCs cultured two-dimensionally and as spheroids.

## 2. Materials and Methods

### 2.1. ADSC Collection and Ethics

In the present study, subcutaneous fat tissue was collected cosmetically from patients who gave informed consent. The study was approved by the Kanazawa Medical University Specified Certified Regenerative Medicine Committee and the Institutional Review Board for Genetic Analysis Research (ID: G129). ADSCs were isolated using an adipose tissue stem cell separation sheet (Bio Future Technologies Co., Ltd., Tokyo, Japan) according to the manufacturer’s instructions. Human ADSCs were cultured using KBM ADSC-1 medium (Kohjin Bio Co., Ltd., Sakado, Japan) and Mesenchymal Stem Cell Growth Medium 2 (MSC-GM2; Takara Bio Inc., Kusatsu, Japan) up to passage 6, with the passage after separation set to 2 [44]. A total of 0.5 g/L-Trypsin/0.53 mmol/L-EDTA Solution (FUJIFILM Wako Pure Chemical Corporation, Osaka, Japan) was used to detach ADSCs. Collected ADSCs met the criteria for ADSCs following analysis using surface markers using fluorescence-activated cell sorting and analysis of the differentiation potential.

The ADSCs used in this experiment were derived from 2 males and 2 females, with a total of 4 samples as follows: ADSC1, 35-year-old male, body mass index (BMI) of 40.1 kg/m^2^, collected from the upper buttocks and right abdomen; ADSC2, 20-year-old female, BMI of 25.5 kg/m^2^, collected from buttocks (including left thigh); ADSC 3, 44-year-old female, BMI of 22.5 kg/m^2^, collected from the lower abdomen (around the buttocks); and ADSC 4, 37-year-old male, BMI of 23.2 kg/m^2^, collected from the abdominal degreasing on the upper and lower sides of the waist [28]. Among the 30 samples banked at our facility, only 2 were male samples; thus, we removed the influence of sex on the experiment by establishing a 1:1 sex ratio.

### 2.2. Spheroid Formation Using SphereRing

ADSCs (required amount ≥4.0 × 10^6^ cells) were cultured according to the above procedure and passed through a Falcon cell strainer (100-µm mesh, Corning Inc., Corning, NY, USA) to obtain a single-cell suspension. A cell suspension was prepared using a cell density of 2.0 × 10^5^ cells/mL and adjusting the total volume to 20 mL using SphereRing (Fukoku Co., Ltd., Ageo, Japan). After injecting 20 mL of cell suspension and 60 mL of air, the SphereRing containing cells was placed on an orbital shaker (OS-762, Optima Inc., Tokyo, Japan) and swirled at a speed of 25 to 65 rpm (Figure 1). The formed spheroids were photographed to measure their size, and their area was measured using ImageJ software (National Institutes of Health, Bethesda, MD, USA).

After culturing the cells, the culture medium was removed from the SphereRing using a syringe and collected into a centrifuge tube. After fixation in 10% neutral buffered formalin solution, the sample was dehydrated by alcohol ascending series (80%, 90%, and 100% three times) and then replaced with xylene and paraffin in that order to make paraffin blocks. After fixing the tissue pieces on glass slides, HE staining was performed, and the stained tissue pieces were observed using a microscope to confirm the state of cells inside the prepared spheroids. Each cell sample was cultured at 37 °C in a humidified atmosphere of 5% CO_2_ in mesenchymal stem cell growth medium 2 (MSC-GM2; Takara Bio Inc., Kusatsu, Japan). The cells reached 80% confluence in the dish and were subcultured.

### 2.3. Extraction of Total RNA from Cultured Cells

Total RNA was extracted from cells cultured in a monolayer (1.0 × 10^4^ cells/mL) 3 days after passage and from spheroids 3 days after the starting of the culture using SphereRing. Monolayer cells were cultured in cell culture dishes (TPP Techno Plastic Products AG, Trasadingen, Switzerland) as controls and compared with cells spheroidized using SphereRing. Total RNA was extracted using RNeasy Mini Kit (Qiagen NV, Venlo, Nederlands) according to the manufacturer’s instructions. The concentration of the extracted RNA was measured using a NanoDrop 2000 (Thermo Fisher Scientific Inc., Waltham, MA, USA). The RNA integrity number value of each sample was measured using an Agilent 2100 Bioanalyzer (Agilent Technologies Inc., Santa Clara, CA, USA).

### 2.4. Transcriptome Analysis Using DNA Microarray

RNA was extracted from cells cultured in a monolayer 3 days after cell passage and from spheroids 3 days after the start of culture using SphereRing. For the DNA microarray analysis, tcDNAs were prepared and applied to the Human Gene ST 2.0 microarray (Affymetrix, Thermo Fisher Scientific, Cleveland, OH, USA) according to the manufacturer’s instructions. The GeneChip WT PLUS Reagent Kit (Affymetrix) was used to synthesize cDNA samples to be hybridized to the microarray probes. For hybridization, we used GeneChip Hybridization Oven 645 (Affymetrix). After hybridization, the microarrays were washed and reacted in a GeneChip Fluidics Station 450 (Affymetrix) using the GeneChip Hybridization, Wash, and Stain Kit (Affymetrix) to colorize them according to the amount of the combined cDNA. The hybridized cDNAs were measured and quantified by a GeneChip Scanner 3000 7G. Data were analyzed using Genespring 14.9.1 (Agilent) and Ingenuity Pathway Analysis (IPA, Qiagen) [43]. Up- or downregulated genes were extracted using the GeneSpring GX software 14.9 (Agilent). IPA (Qiagen, Hilden, Germany) was used for gene network evaluation [45,46].

### 2.5. cDNA Synthesis and Polymerase Chain Reaction (PCR) Analysis

The cDNA was synthesized using the SuperScriptIII First-Strand Synthesis System for RT-PCR Kit (Thermo Fisher Scientific) according to the manufacturer’s instructions. PCR was performed on the obtained cDNA using TaKaRa Ex Taq (Takara Bio) and the PCR primers shown in Table 1. The thermal cycler conditions were as follows: 35 cycles of 10 s at 98 °C, 30 s at 56 °C, and 60 s at 72 °C. PCR products were analyzed by 1.5% agarose gel electrophoresis and photographed using an e-gel imager (Life Technologies, Carlsbad, CA, USA).

### 2.6. SF Collection and Measurement of Viable ADSCs

SF was collected from the knee joint of OA patients (Kellgren–Laurence classification grade 3) who gave informed consent under the approval of the Kanazawa Medical University Medical Ethics Review Board Approval number: I583 [47,48].

Because the use of analgesics drugs and NSAIDs is known to alter the immunomodulatory factors of SF [49], SF collected from untreated knee OA patients were used in this study.

After disinfection of the knee area with povidone-iodine, the SF was collected by joint aspiration using an 18G needle with a superolateral approach under ultrasound guidance [48]. The collected SF was immediately centrifuged following cryopreservation at −80 °C. SF was thawed at room temperature for use just before the experiments.

MSC-GM2 medium was used as a control, and ADSCs were counted in a two-dimensional monolayer culture in MSC-GM2 and added at an arbitrary concentration of 50 μL per well of a 96-well plate. After incubating for 48 h (37 °C, 5% CO_2_), 50 μL of 0% (control), 10%, 20%, and 40% SF was added to each well. Spheroids were prepared, then seeded in a 96-well plate in an arbitrary amount, and SF was added to each well. After incubating the plates for 4 and 16 h, the relative light units (RLUs) were measured in each well using a CellTiter-Glo 3D Cell Viability Assay (Promega Corp., Madison, WI, USA) according to the manufacturer’s protocol using a GloMax 96 Microplate Luminometer (Promega). This kit enables the measurement of the viability of three-dimension cultured cells. The percentage of RLU was measured by subtracting the RLU of the culture medium and SF containing no ADSCs.

### 2.7. Comprehensive Cytokine Assay

Our ADSC samples were used as described above. The cytokine concentrations in the spheroid culture medium made using SphereRing and in the supernatant of 2D monolayer cultured cells were comprehensively measured to compare the 24 h cytokine production per cell. For spheroids, 4.0 × 10^6^ cells/20 mL ADSC suspension and 20 mL of MSC-GM2, respectively, were administered to SphereRing and cultured in an orbital shaker (35 rpm, 37 °C, 5% CO_2_) for 48 h. Monolayer culture cells were seeded in 6-cm dishes with 5 mL of MSC-GM2 (Takara Bio Inc., Kusatsu, Japan) containing 1.0 × 10^4^ cells/mL ADSCs and incubated at 37 °C and 5% CO_2_ for 48 h. Both spheroid and monolayer cells were incubated for 24 h. The culture medium was changed 48 h after the starting of the incubation. The cells were cultured for 24 h. The culture supernatant was then collected, centrifuged, and immediately stored at −80 °C. The culture supernatant was subjected to Bio-plex (27-plex panel, Bio-Rad Laboratories, Hercules, CA, USA). The following cytokines were measured: interleukin-1b, 1ra, 2, 4, 5, 6, 7, 8, 9, 10, 12, 13, 15, and 17, as well as granulocyte-colony stimulating factor (G-CSF), granulocyte-macrophage colony-stimulating factor (GM-CSF), interferon-gamma (INF-γ), tumor necrosis factor-α (TNF-α), monocyte chemoattractant protein-1 (MCP-1), CXCL10 chemokine (IP-10), MIP-1a, MIP-1b, RANTES, eotaxin, platelet-derived growth factor (PDGF), basic-fibroblastic growth factor (bFGF), and vascular endothelial growth factor (VEGF). All sample measurements were performed in duplicates according to the manufacturer’s protocol. If the measured value was below the limit of detection (LOD) for a cytokine, the estimated value was calculated using the respective LOD. The absorbance of each sample was measured by the Cell Titer-Glo 3D Cell Viability Assay (Promega Corp., Madison, WI, USA), and the cytokine production quantity per 24 h per cell was calculated by estimating the 24 h cytokine release per cell.

### 2.8. Statistical Analysis

Data in this study are presented as mean ± SD or SE. Statistical analysis of the means between the two groups was performed using Student’s *t*-test; *p* < 0.05 was considered statistically significant. Statistical analysis was performed using IBM SPSS Statistics version 27. Fisher’s exact probability test was used to examine the association of IPA involvement in the extracted variation genes.

## 3. Results

### 3.1. Culture of ADSCs Using SphereRing

The optimal conditions for ADSC spheroid formation using SphereRing were investigated using various swirling speeds. The state of the cultured spheroids was observed under a phase-contrast microscope following swirling speeds of 25, 35, 45, 55, and 65 rpm, which led to successful spheroid formation. Images of typical cultured cells under a phase-contrast microscope are shown in Figure 2.

Uniform and large spheroids were observed under the condition of 35 rpm. At 25 rpm, adhesion of a large number of cells was found attached to the inside of the SphereRing tube. The area was also measured using images taken under a phase-contrast microscope at each swirl speed on day 3 of culture. Changes in the size of spheroids at each swirl speed are shown in Figure 3. The inhibition of cell aggregation associated with the swirling speed increased more than the force with which the cells adhered to each other.

### 3.2. Pathological Specimens of the Spheroids

Pathological specimens of the spheroids on day 3 of culture were prepared to confirm the state of cells inside the cultured spheroids (Figure 4). We observed an internal cavity in a part of the prepared spheroids. We considered it a gap between the cells that formed when agglomerates adhered to each other during spheroid formation. In the swirling cultures, in which individual populations repeatedly adhered to each other in a stream of medium, we suggest that spheroids might not exist as uniform aggregates in the internal tissues. However, the inner spheroid cells with a high cell density were not largely necrotic. Therefore, aggregates could be formed under the culture conditions of the present study while maintaining the state of living cells.

### 3.3. Transcriptome Analysis Using DNA Microarray

The characteristics of ADSCs were examined at the gene expression level in spheroids compared with cells cultured in a two-dimensional monolayer. Total RNA was extracted, and fluctuations in gene expression were comprehensively analyzed using DNA microarray. This study is the first experiment that used SphereRing to form spheroids using human ADSCs. We performed a series of DNA microarray analyses to comprehensively reveal what events are occurring compared to normal 2D cultured cells.

Among ADSC1 and ADSC2, which were prepared and analyzed independently, 39 genes were upregulated, and 41 genes were downregulated (FC4.0). The number of genes that fluctuated in common in ADSC1 and ADSC2 is shown in Figure 5c. Gene ontology analysis using IPA was performed using the common fluctuating gene cluster to clarify the outline of the fluctuation in expression. Table 2 and Table 3 show the main gene categories that changed. The *p*-value was calculated using Fisher’s exact probability test in this study. The smaller the *p*-value, the less likely that the association is random. The term #Molecules represented the number of molecules that fit into the GO term.

As a result of Gene ontology analysis revealed that most of the genes that fluctuated in common were related to the cell cycle, cell migration, adhesion, and proliferation. Cell cycle-related gene clusters were confirmed in the gene clusters that fluctuated in common, and it is possible that proliferation was suppressed. In addition, cell-to-cell signaling and the interaction category were confirmed, which may have contributed to cellular adherence. IPA also indicated that the gene network included Akt, Mapk, and PI3K (Figure 6). These gene network changes were considered unusual under spheroidal conditions.

### 3.4. Semi-Quantitative RT-PCR

DNA microarray analysis revealed that the expression of many genes fluctuated. Therefore, we performed semi-quantitative RT-PCR to validate our DNA microarray results and confirm whether similar gene expression fluctuations were observed in the ADSCs from the four patients (Figure 7).

In this experiment, RT-PCR was performed for the genes *TNC*, *COL15A*, and *ANGPTL2*, which showed increased expression with microarray analysis, as well as *KRT34* and *KRTAP2–3*/*2–4*, which showed decreased expression. *COL15A1* and *ANGPTL2* are associated with vascular endothelial cells and angiogenesis [50], whereas *TNC* is associated with tissue development [51]. On the other hand, *KRT34* and *KRTAP2–3/2–4* are involved in the fiber structure of hair [52]. Similar to the findings reported by Cesarz et al., the expression of genes related to angiogenesis was increased [30]. This suggests that the spheroids generated using the SphereRing in the present study showed the same properties as those produced using existing methods.

### 3.5. Comprehensive Cytokine Assay

In order to further evaluate the characteristics of the spheroids prepared using the SphereRing, we compared the production of cytokines by multiplex assay, and as shown in Figure 8, there was an increase in the production of many cytokines in the culture supernatant of the spheroids compared to 2D-cultured cells. IL-6, IL-9, and MCP-1 could also be detected in the supernatant of 2D culture at concentrations above the detection limit, but spheroids were detected to accumulate at significantly higher concentrations. On the other hand, the concentrations of IL-5, IL-6, IL-9, IL-10, Eotaxin, MCP-1/MCAF, MIP-1β, and VEGF were below the detection limit in the supernatant of 2D-cultured cells. Therefore, a significant increase in accumulation was observed for spheroids in the test using the detection limit as the measurement value.

### 3.6. Effect of SF on Cultured ADSCs

Intra-articular administration of ADSCs is known to cause cell death within days to weeks [53]. However, few studies report on how intra-articular SF affects therapeutically administered ADSCs. Therefore, SF was collected and added to ADSCs in culture to examine whether it was toxic to spheroids formed by ADSCs. Although it is preferable to collect ADSCs and SF in the same patient, in clinical practice, OA treatment with ADSCs is often already performed with therapeutic intervention. In this study, we did not compare the variability of fluid factors in the same patients because of the possibility of not being able to eliminate the influence of therapeutic intervention.

SF collected for therapeutic purposes from patients with OA was added to the culture medium at varying concentrations (0%, 10%, 20%, and 40%), and the rate of cell death was measured after culturing for 4 and 16 h. The two-dimensional culture of ADSCs was examined (Figure 9a) and showed that the addition of SF significantly induced cell death as previously reported. SF was also added to spheroids formed using SphereRing (Figure 9b), and the rate of cell death was measured. Some dead cells were also observed in the spheroid culture. The increase in %RLU observed between 4 and 16 h may have been due to the proliferation of ADSCs.

## 4. Discussion

Multiple studies reported the effects of spheroids formed with ADSCs; Feng et al. reported high therapeutic efficacy in wound repair in vivo along with excellent stem cell properties in vitro [37]. Zhang et al. showed that spheroid formation activates anti-inflammatory, anti-apoptotic, and angiogenic effects, as well as homing to the damaged site [35]. Ko et al. compared ADSC spheroid and single-cell suspension viability and cartilage regeneration in an OA mouse model; the ADSC spheroid-treated group showed improved intra-articular cell viability and a significant OARSI score improvement, indicating the superiority of spheroids [54].

The present study is the first to report successful spheroid formation using human ADSCs with SphereRing. We first examined and evaluated the conditions for producing cellular spheroids and found that lower turning speeds promoted spheroid formation. This was observed in images obtained during culture and by calculating the spheroid area (Figure 2 and Figure 3). This likely occurred because the centrifugal force due to swirling is a major factor in the swirling culture, in which adhesion is repeated in a water stream to form aggregates. Centrifugal force is higher with higher turning speeds, and conversely, lower turning speeds result in a weaker centrifugal force, which is thought to inhibit cell aggregation. On the other hand, although larger spheroid formation was observed at 25 rpm, the formed aggregates did not float and adhered to the bottom surface. Therefore, a surface coating that suppresses cell adhesion may be effective to improve SphereRing^®^ spheroid formation [55]. In addition, if the spheroid is too large, the culture medium cannot reach the inside and necrosis might occur. Moreover, large spheroids could cause clogging of the injection needle when administered into the joint cavity. Therefore, we deemed that the optimal swirling speed was 35 rpm for spheroid formation for further studies.

DNA microarray transcriptome analysis of monolayer-cultured cells grown on flat dishes and spheroids cultured using SphereRing showed that the expression of several genes changed when cells were cultured three-dimensionally (Figure 5 and Figure 6, Table 3). Validation with RT-PCR also supported the results of the DNA microarray analysis (Figure 7). This is likely because the target of cell adhesion was changed from a plastic substrate to cell-to-cell. Since the stiffness and cell density of the adhesion target changed greatly between the dish and the cells, these factors may be linked to changes in gene expression. In addition, the function of the variable genes extracted by the gene analysis software IPA showed changes in genes related to cell cycle and cell motility. Genes related to cell–cell interaction, cell assembly, and organization also showed significant changes, suggesting that the formation of dense cell–cell aggregates led to changes in genes related to active cell–cell signaling and interaction. The mechanism of action of ADSCs in joints may be explained by angiogenic factors, immunosuppressive factors, and tissue regeneration [56], and the effect may be enhanced by spheroid formation. IPA revealed a gene network including Akt, MAPK, and PI3K, and MSC-derived spheroids reportedly induce PI3K-Akt signaling pathway activation, which might be similar to what occurred in the present study [57].

Yoon et al. found that spheroid-cultured ADSC implantation into mouse knee joints promoted in vivo chondrogenesis compared with the implantation of monolayer-cultured ADSCs [58]. The study implicated hypoxia-induced cascades and cell–cell reinforcement, suggesting that spheroid formation-induced hypoxia might favor cartilage regeneration. In this study, cell death and survival were also variable in the GO analysis, and we speculated that they might be involved in the cartilage regeneration mechanism. From the ADSC spheroid microarray analysis results, genes involved in cartilage regeneration and formation were identified, including *SPON1* [59], *PDGF* [60], *COL10A* [61]. Moreover, other gene variations involved in cartilage regeneration and formation were confirmed. This revealed that the spheroids created by SphereRing might have also acquired chondrogenic differentiation ability.

Cytokine assays performed to characterize the spheroid showed a significant difference in cytokine production compared to the 2D-cultured cell supernatants for eotaxin, IL-5, IL-6, IL-9, IL-10.MCP-1/MCAF.MIP-1β, and VEGF (Figure 8). Increased cytokine secretion from the supernatant has been reported when ADSCs are used to make spheroids [62,63], and we confirmed similar results in this study. It is possible that this cytokinesis increase is due to the hypoxic state of the spheroid inner core. In this study, it was difficult to isolate the spheroid and measure the cell numbers, so we decided to use a 3D assay to approximate the number of cells. Calibration curve-based calculation might result in lower detection of spheroid-produced cytokines, but we considered this to be within an acceptable range. The microarray analysis results did not show any difference in the cytokine expression between spheroids and monolayer cells (data not shown). The cytokine production increase in spheroids was not at the RNA level, but rather the protein synthesis and release after translation. The increased cytokine release compared to 2D-cultured cells is one of the main spheroid features and is expected to be a useful method for intra-articular administration.

In clinical practice, when ADSCs are cultured and administered to patients with knee OA, it is common to administer them into the joint cavity. In this case, it is inevitable that administered ADSCs would come into contact with the SF in the joint cavity. The SF reportedly exerts a cytotoxic effect on canine ADSCs [43], but few studies reported on the effect of SF on human ADSCs. This is an important issue when considering the clinical application of spheroid ASDCs. In this study, we compared the resistance of cultured ADSCs to the SF between 2D and spheroid cells (Figure 9). We observed no significant difference between them, but the SF exhibited a lethal effect on the ADSCs. To the best of our knowledge, this is the first report of such an effect on human ADSCs. In order to clarify the usefulness of spheroids in the treatment of OA, it is necessary to set up experimental conditions that are more suitable for the environment in the joint cavity and to study a larger number of cases. In the future, the cytotoxic activity of the SF against ADSCs administered into the joint cavity for therapy should be considered when improving culture methods.

We believe that the relationship between ADSCs and the SF cannot be ignored as intra-articular ADSC administration becomes a common treatment in the future. It is necessary to control the effects of synovial fluid to improve treatment outcomes. In the future, the SphereRing can be used to produce relatively large spheroids, and it is expected to be used in medium-sized animals such as pigs that are similar to humans in several aspects. SphereRing can produce a large number of spheroids at a time at a low cost, and it is easy to recover the spheroids, as they are concentrated by precipitation if the culture solution is left to stand. In addition, we observed that SphereRing enhanced cytokine production and useful gene expression compared to the 2D-cultured cells; therefore, ADSC spheroid creation using SphereRing could be a very useful tool for ADSC therapy. In the future, we hope to contribute to ADSC spheroid creation that is more viable in the knee joint and thus more effective in treatment.

## Figures and Tables

**Figure 1 cells-11-00337-f001:**
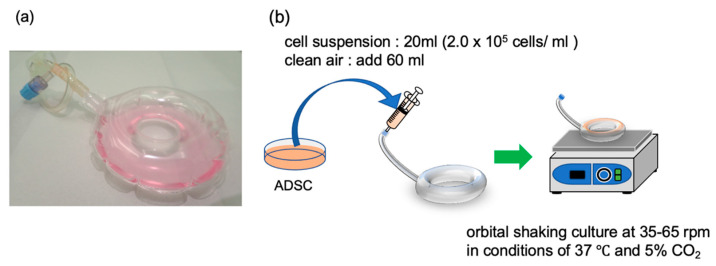
SphereRing and schematic of the culture method. (**a**) Image of the SphereRing. (**b**) Procedure for forming spheroids using SphereRing on a shaker.

**Figure 2 cells-11-00337-f002:**
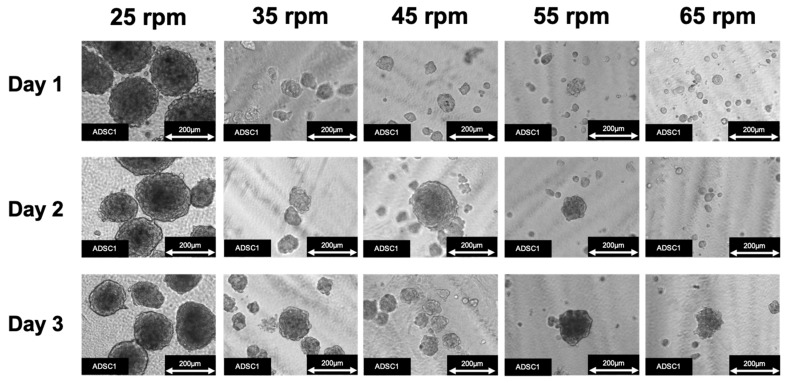
ADSC spheroid formation using SphereRing at different swirling speeds.

**Figure 3 cells-11-00337-f003:**
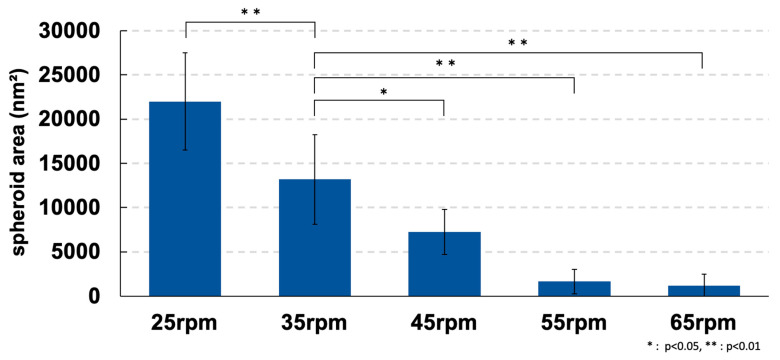
Spheroid area for each condition on day 3 of culture. Data are expressed as the mean ± SD. * shows *p* < 0.05 and ** indicates *p* < 0.01.

**Figure 4 cells-11-00337-f004:**
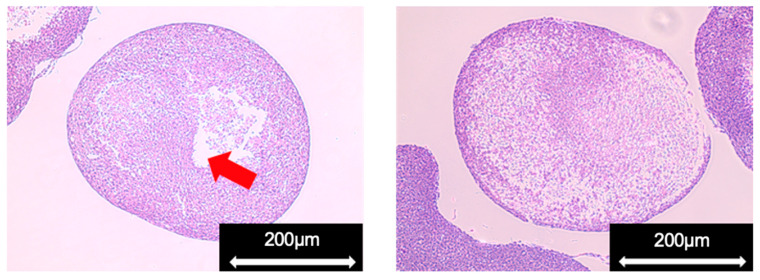
Pathological specimen of a spheroid (ADSC1). Red arrow indicates cavities in the spheroid.

**Figure 5 cells-11-00337-f005:**
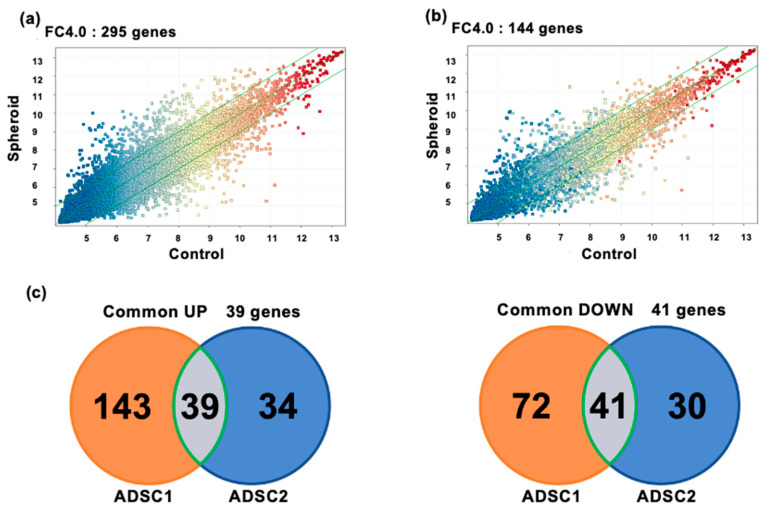
Transcriptome analysis using DNA microarrays. (**a**) Scatter plots of DNA microarray analysis of ADSC1. (**b**) Result of DNA microarray analysis of ADSC2. (**c**) Venn diagram comparing gene expression in spheroid-cultured cells and 2D monolayer-cultured cells showing the number of genes whose expression increased or decreased and the genes that commonly changed in ADSC1 and ADSC2. Common UP shows the number of genes whose expression increased, and Common DOWN shows the number of genes whose expression decreased. In common UP (orange circles), 182 genes were upregulated in both spheroid-cultured and 2D monolayer-cultured ADSC1. In ADSC2, 73 genes were upregulated in both spheroid culture and 2D monolayer culture (blue circles). There were 39 genes that were upregulated in both ADSC1 and ADSC2 (gray circles). In common DOWN, 113 genes in ADSC1 were downregulated in both spheroid and 2D monolayer cultures (orange circle). A total of 71 genes were downregulated in ADSC2 in both spheroid culture and 2D monolayer culture (blue circles). A total of 41 genes were downregulated in both ADSC1 and ADSC2 (gray circles).

**Figure 6 cells-11-00337-f006:**
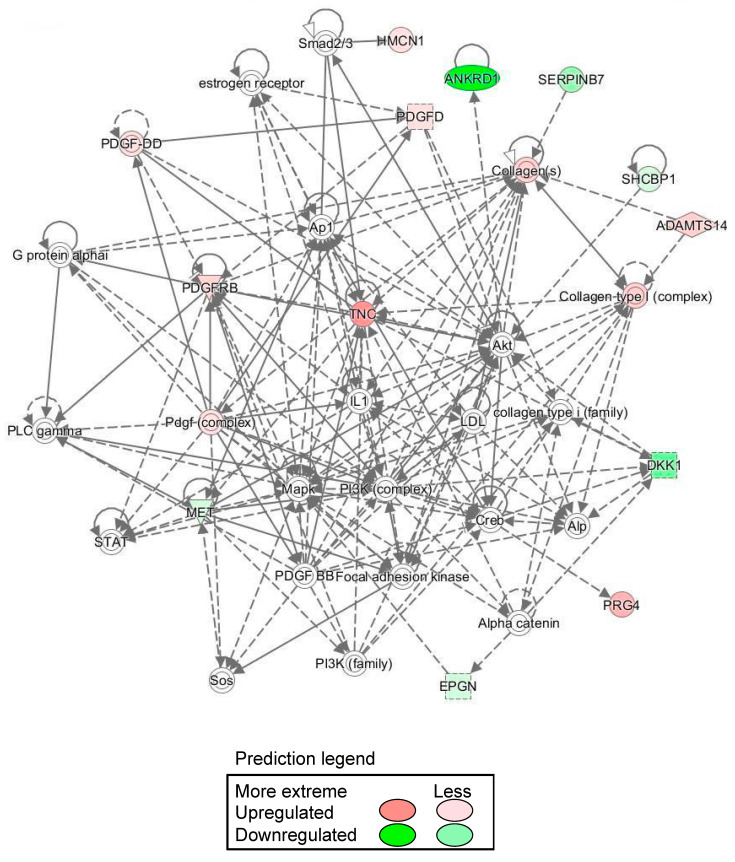
Ingenuity pathway analysis-identified spheroid gene network. IPA also predicted the gene network included Akt, MAPK and PI3K.

**Figure 7 cells-11-00337-f007:**
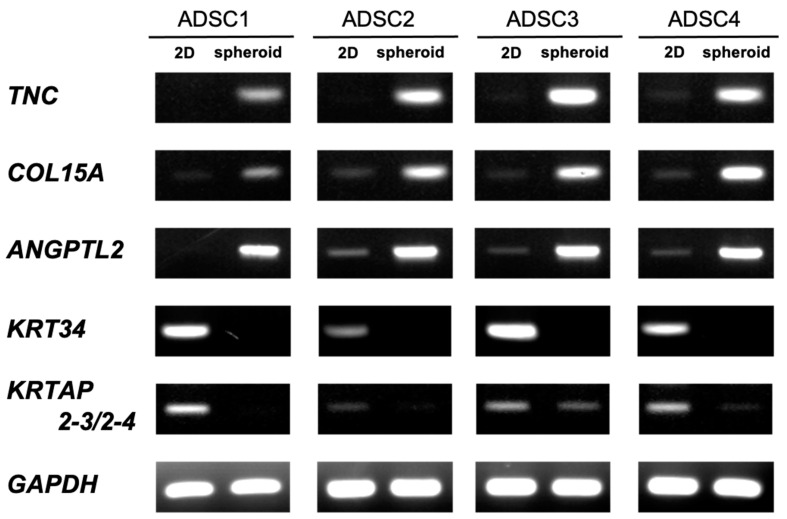
Validation of mRNA expression of screened genes by semi-quantitative RT–PCR for ADSCs.

**Figure 8 cells-11-00337-f008:**
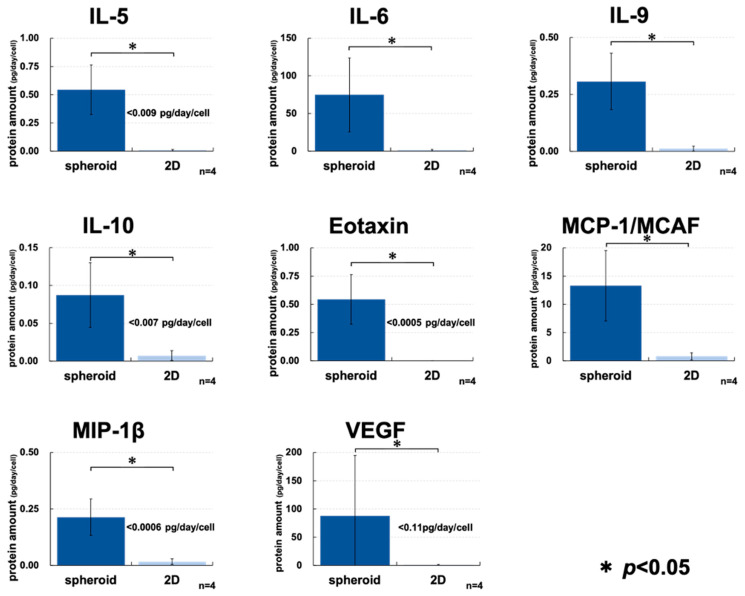
ADSC cytokine release in the spheroid and 2D monolayer culture supernatants. The spheroid and 2D monolayer cultures were incubated for 48 h, and the supernatants were collected and measured 24 h after the culture medium was changed. To estimate the number of cells in the spheroid and monolayer culture cells, the number of cells was measured by comparing the amount of luminescence by the Cell Titer-Glo 3D Cell Viability Assay. The number of cells was used to calculate the amount of cytokines produced per cell after 24 h. When a measured value below the detection limit was obtained, the detection limit was indicated in the graph. In addition, a statistical test was performed using the detection limit as the measured value. * indicates *p* < 0.05 using Student’s *t*-test.

**Figure 9 cells-11-00337-f009:**
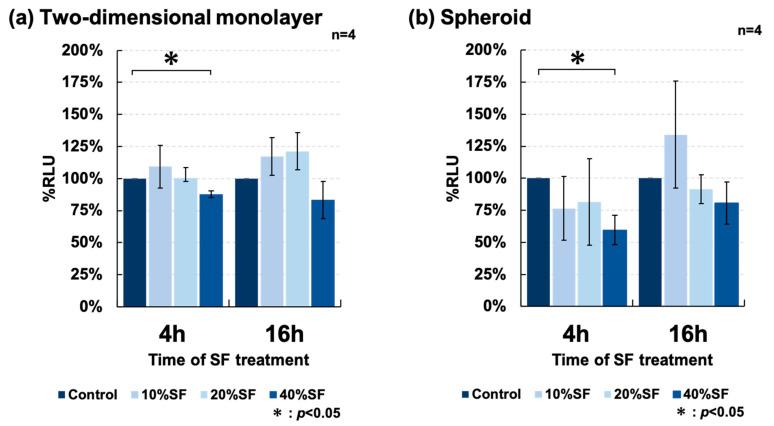
Cell survival rate of ADSCs cultured with synovial fluid (SF). (**a**) The effects of synovial fluid (SF) on cell viability of ADSCs cultured as (**a**) two-dimensional monolayer and (**b**) spheroids after 4 and 16 h of culture. Data are expressed as the mean ± SE.

**Table 1 cells-11-00337-t001:** Primers and product sizes.

Gene Symbol	Primer	Sequence (5′–3′)	Product Size
*GADPH*	F	CAACGAATTTGGCTACAGCA	195
	R	AGGGGTCTACATGGCAACTG	
*TNC-R*	F	GGTACAGTGGGACAGCAGGT	279
	R	GCCTGCCTTCAAGATTTCTG	
*COL15A1*	F	GCTTTGGCTTTTGAGTCCAG	293
	R	AGGATGGAGTTGGAGGTGTG	
*ANGPTL2*	F	CTGGGCCTGGAGAACATTTA	334
	R	CTCGGAACTCAGCCCAGTAG	
*KRT34*	F	GAGCTGACCCTCTGCAAGTC	292
	R	GCTGCTCTGAGCTGGATACC	
*KRTAP2–3/2–4*	F	CTTGTCCTCCCTGAGCTACG	292
	R	GGGACTGCACAGACACAGG	

**Table 2 cells-11-00337-t002:** Gene ontology analysis of extracted upregulated genes.

Molecular and Cellular Functions		
GO term	*p*-value	#Molecules
Cellular Movement	3.02 × 10^3^–3.39 × 10^5^	9
Cell Morphology	3.02 × 10^2^–2.50 × 10^4^	7
Cellular Development	2.19 × 10^2^–2.50 × 10^4^	15
Cell Death and Survival	2.69 × 10^2^–1.70 × 10^3^	7
Cell-To-Cell Signaling and Interaction	2.86 × 10^2^–1.70 × 10^3^	8

**Table 3 cells-11-00337-t003:** Gene ontology analysis of extracted downregulated genes.

Molecular and Cellular Functions		
GO term	*p*-value	#Molecules
Cell Cycle	4.32 × 10^3^–6.55 × 10^7^	18
Cellular Assembly and Organization	4.32 × 10^3^–2.48 × 10^6^	13
DNA Replication, Recombination, and Repair	4.71 × 10^3^–2.48 × 10^6^	6
Cell Death and Survival	5.65 × 10^3^–3.80 × 10^6^	16
Cellular Movement	5.46 × 10^3^–5.00 × 10^6^	13

## Data Availability

The authors declare that the data supporting the findings of this study are available on request.

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
