# Peer review of "Evaluation of the Usefulness of Human Adipose-Derived Stem Cell Spheroids Formed Using SphereRing® and the Lethal Damage Sensitivity to Synovial Fluid In Vitro"

_cells, 2022, doi:10.3390/cells11030337_

Round 1

Reviewer 1 Report

Dear Authors, the article now appears more consistent and clear, however few points remain to be clarified, as explained below.

Abstract section, line 46: it is non elegant to list some cytokines and write etc, instead you can write only “pro or anti-inflammatory cytokines and growth factors” or only “cytokines and growth factors”.

Line 520. Please insert what is your idea regarding cellular death following contact between spheroids and synovial fluids. Wouldn't a protective result of the spheroids be expected with respect to the 2d culture?

Author Response

Thank you very much for your kind advice.

About line520, We had hoped that spheroid would have a protective effect, but there was no significant difference at this point.

however, to the best of our knowledge, this is the first study to confirm a lethal effect in humans.

It is thought that it is necessary to set up experimental conditions that are more suitable for the environment in the joint cavity and to increase the number of cases for further study.

In the future, the cytotoxic activity of SFs against ADSCs administered into the joint cavity for OA treatment should be taken into account when improving the culture method.

Therefore “In order to clarify the usefulness of spheroids in the treatment of OA, it is necessary to set up experimental conditions that are more suitable for the environment in the joint cavity and to study a larger number of cases.” We  have added and shown in red. (lines 522-525)

Reviewer 2 Report

The authors of the study have substantially improved the manuscript from the original submission. The authors need to improve the following points prior to further manuscript consideration,

  1. Correct VGEF to VEGF
  2. Improve the presentations of the graphs in figure 3, Figure 8 and Figure 9. These should be consistent and use of either Graphpad or SPSS is advised.

Author Response

Thank you very much for your precious feedback.

I will correct it from VGEF to VEGF in Figure8 and line91 as you commented.

As for the graphs that you pointed out, we have made them again using SPSS version 27. Figure3, 8, and 9 have all been corrected to be more consistent. In addition, we added a graph scale to make it easier for the reader to understand.
 We added the use of SPSS in 2.8. Statistical analysis (line290-291).

Reviewer 3 Report

I accept the authors' revisions. 

Author Response

I deeply appreciate your decision. Thank you very much.

This manuscript is a resubmission of an earlier submission. The following is a list of the peer review reports and author responses from that submission.

Round 1

Reviewer 1 Report

The innovative part of this work is the production of the spheroids to be used for the treatment of OA, but the authors failed to clearly make a case in favor of the use of these spheroids vs that of individual AD-MSCs by comparing the effects of both treatments in the introduction. Also, what is the real purpose of these spheroids? Direct tissue regeneration or enhancing the potential of endogenous MSCs?

Even if the cells were grown under hypoxic conditions (something that is not specified), considering that MSCs rapidly enter replicative senescence, passage 6 seems like a late passage to use for these experiments. This is worrying since the potential of the cells after they have been cultured and spheroids has not been tested. Are those MSCs forming the spheroids capable of chondrogenic differentiation? A gene expression analysis would simply not do it. Considering the complexity of the chondral tissue, the authors should do specific histological analysis of the glycosaminoglycans content after differentiation on top of an analysis of the expression of key chondrogenic markers to prove that those cells keep their potential after the manipulation.

On the other hand, if the spheroids intend to be used to reduce inflammation, are they able to produce anti-inflammatory cytokines in culture?

These kind of  tests are needed to put a case in favor of the effectivity of those spheroids and thus, of its application in the clinical practice.

All in all, in this reviewer´s opinion, this manuscript does not provide sufficient advance in the field to grant its publication in Cells. The article does not provide information that could be useful to translate the results to the clinic or to assay the potency of the spheroids and the only part this reviewer finds innovative is the description of the culture method using the SphereRing. It would therefore be premature to proceed further with the manuscript on the basis of the present results. Authors should demonstrate that the spheroids formed using this method have the characteristics or properties attributed to these structures, in vitro as well as in animal models.

Regarding the writting, there is a high number of mistakes such as duplicate and missed words, unfinished or meaningless sentences, spelling mistakes, etc. that do feel as if the manuscript was produced in a hurry, and sometimes makes it difficult to understand what the authors mean.

Some conceptual mistakes are also found, such as the designation of the platelet rich plasma as a “cell source”.

Reviewer 2 Report

The authors characterised adipose derived stem cell spheroids manufactured using a SphereRing® system. PCR analysis was performed on spheroids and synovial fluid from patients was collected to check the spheroid viability. Extensive gene ontology analysis is shown, with TNC, COL15A1, ANGPTL2 having increased expression within the spheroid. In presence of synovial fluid from OA patients, there is increased cell death with higher concentrations of synovial fluid, although not significant compared to the control.

The manuscript characterizes the spheroids and provides interesting microarray data.

The authors should answer the following questions

  1. The applications of these spheroids for the treatment of OA has been postulated to be related to secretomic factors. Thus, what the secretomic factors produced from these spheroids ? The authors should present data on the secretomic factors released into media to show that it would induce regeneration or produce anti-apoptotic factors for the treatment of early OA lesions ELISA analysis of specific growth factors would be sufficient to answer this.
  2. Do the authors have data regarding either chondrocyte redifferentiation or MSC chondrogenesis to prove that these spheroids would induce cartilage regeneration ? Evidence that these spheroids stimulate chondrogenic genes is required to prove its efficacy in treating early OA.
  3. What was the rationale for performing an extensive microarray analysis rather than a simple experiment that demonstrated the efficacy of the spheroid application for chondrocyte or MSC chondrogenesis ? The authors should have a clear justification for this purpose.
  4. For clinical application, how would these spheroids be used to treat patients with OA ? Authors should write about how the spheroids would be delivered to the joint (i.e. carrier hydrogel or direct injection without a biomaterial) for their purpose.

Reviewer 3 Report

The paper on “Evaluation of the usefulness of adipose-derived stem cell (ADSC) spheroids formed using SphereRing® in knee osteoarthritis.” reported on interesting data regarding the usefulness of adipose-derived stem cells and newly developed spheroid.

The study was well designed. Also, the paper was nicely written and the authors used a good range of modern technologies to corroborate their hypothesis.

However, there are several critical points, which need to be addressed or revised before acceptance.

Comments.

  1. In title, the authors demonstrated the usefulness of adipose-derived stem cell spheroids formed using SphereRing® in knee osteoarthritis (OA). I wonder where in the ADSC-spheroids useful is for knee OA? Indeed, the authors mentioned the usefulness of ADSCs and spheroids for articular cartilage repair. However, they did not show the data regarding the effect of ADSC-spheroids formed using SphereRing®. I accept the usefulness of ADSCs in the tissue engineering. Of course, I understand the necessity of spheroid in tissue repair using mesenchymal stem cells. Maybe, ADSC-spheroids formed using SphereRing® might have a potential for the cartilage repair. However, the authors should verify the issue in their experiment. Did the authors have any data about the usefulness of it in the cartilage tissue engineering?
  2. As the authors also demonstrated in the discussion section, numerous reports have already demonstrated the effects of spheroid formed using ADSCs. How about the differences between the SphereRing® and other spheroids?
  3. Regarding the contact of ADSCs with synovial fluid, did they derive from same donor?

In my opinion, these comments and questions are thought to be critical points for journal readers.

Reviewer 4 Report

Although this study may appear interesting at times, it lacks some information while others are unclear. Importantly, the English used is really poor and must absolutely be revised.

The ADSC and the synovial fluid derived from the same patient with Knee OA?

How the authors justify the choice of patients who are so heterogeneous in term of age, gender and BMI?

Why the adipose tissue derives from different sampling areas? It is known from the literature, how these factors influence the ADSCs.

Figure 5. please add in the legend the meaning of the different colors represented. In addition, the content of the yellow box is illegible. Please rewrite at a higher resolution.

Table 2. please accurately describe the content of this table. What is the mean of “#Molecules”? Where do the indicated p values ​​come from?

Line 36: angiogenesis is repeated twice. Please remove it.

Line 137: the authors report that ASCs derived from 2 males and 2 females, but the description of the patients report one man and 3 females. Please correct it. 

Line 170: please specify the growing conditions (cell/cm2, number of cm2 seeded).

Lines 179-180: please insert “RNA was extracted from cells cultured in monolayer 3 days after cell passage and from spheroids 3 days after the start of culture using SphereRing®” in the previous “2.4. Extraction of total RNA from cultured cells” section.

From line 182 to line 184: this part of the text is poorly written and unclear. It is only a list of actions not well described. Please rewrite it in a correct english form.

Line 192: please change “degree” with °C.

Line 199: please specify if the synovial fluid comes from the same patients from whom the ASCs were taken. If not, describe how many patients the synovial fluids are derived from, if it is a result of a pool or not.

Line 204: the MSC-GM cells are ADSCs in control medium? this wording generates a lot of confusion, please rephrase it more clearly, for example by simply writing "ADSC in control medium".

From line 254  to line 261: this part of the text is poorly written and unclear. Please rewrite it in a correct english form.

The discussion section appears little argued. DNA microarray results are too little discussed and nothing is mentioned on the genes chosen for the validation of the array, in term of their role, the reason for their selection and of the result obtained. Please implement this part.